# Development and Optimization of Lipase-Catalyzed Synthesis of Phospholipids Containing 3,4-Dimethoxycinnamic Acid by Response Surface Methodology

**Magdalena Rychlicka \*, Natalia Niezgoda and Anna Gliszczyńska \***

Department of Chemistry, Wrocław University of Environmental and Life Sciences, Norwida 25, 50-375 Wrocław, Poland; natalia.niezgoda@upwr.edu.pl

\* Correspondence: rychlicka.magda@wp.pl (M.R.); anna.gliszczynska@wp.pl (A.G.); Tel.: +48-71-320-5183 (M.R.)

**Abstract:** The interesterification reaction of egg-yolk phosphatidylcholine (PC) with ethyl ester of 3,4-dimethoxycinnamic acid (E3,4DMCA) catalyzed by Novozym 435 in hexane as a reaction medium was shown to be an effective method for the synthesis of corresponding structured *O*-methylated phenophospholipids. The effects of substrate molar ratios, time of the reaction and enzyme load on the process of incorporation of 3,4DMCA into PC were evaluated by using the experimental factorial design of three factors and three levels. The results showed that a substrate molar ratio is a crucial variable for the maximization of the synthesis of 3,4-dimethoxycinnamoylated phospholipids. Under optimized parameters of 1/10 substrate molar ratio PC/E3,4DMCA, enzyme load 30% (w/w), hexane as a medium and incubation time of 3 days, the incorporation of aromatic acid into phospholipid fraction reached 21 mol%. The modified phosphatidylcholine (3,4DMCA-PC) and modified lysophosphatidylcholine (3,4DMCA-LPC) were obtained in isolated yields of 3.5% and 27.5% (w/w), respectively. The developed method of phosphatidylcholine interesterification is the first described in the literature dealing with 3,4DMCA and allows us to obtain new *O*-methylated phenophospholipids with potential applications as food additives or nutraceuticals with pro-health activity.

**Keywords:** phosphatidylcholine; 3,4-dimethoxycinnamic acid; enzymatic interesterification; lipase; biocatalysis

## 1. Introduction

Methoxylated derivatives of cinnamic acid (CA) are constituents of plants which have immense therapeutic potential. They are known in the literature as biomolecules that exhibit antioxidant properties [1,2] and are active as anti-cancer [3,4], anti-diabetic [5–8], hepato- [9,10] and neuroprotective [11–13] agents. It has been proven that the dietary intake of these compounds has a health-promoting effect on the body and effectively supports the prevention of cancer and metabolic or cardiovascular diseases.

Belonging to the group of natural methoxylated derivatives of CA, 3,4-dimethoxycinnamic acid (3,4DMCA) is an ingredient of fruits, spices and herbs [14]. The richest sources of 3,4DMCA are blueberries (725 mg/kg dm), blackberries (501 mg/kg dm) [15] and coffee (*Coffea canephora* var. Robusta) (690 mg/kg dm) [16], where it occurs in two forms, as a free acid and conjugates with quinic acid.

A diet enriched with 3,4DMCA has been highly recommended for patients with cancer of the stomach, prostate and thymus [3]. Watanabe et al., confirmed that injections of 3,4DMCA in saline solution at 0.25 mmol/kg significantly reduce the concentration of polyamines such as putrescine,



spermidine and spermine, in the in vivo tests, carried out on a rat model. Its high level is observed in the patient's body under tumorigenesis and after chemotherapy [3]. It has been proven that this natural dimethoxycinnamic acid decreases the serum cholesterol without affecting the concentration of high-density lipoprotein (HDL)-cholesterol exhibiting activity comparable to activity of statins [17]. Based on other study, it was also suggested that 3,4DMCA could be an effective dietary compound in prophylaxis of neurodegenerative diseases, since it was discovered that it is able to act as an antiprion factor [18].

The pharmacokinetics and bioavailability of 3,4DMCA acid have been well evaluated on a group of volunteers whose diet, 24 h before coffee consumption, was low in polyphenolic compounds [19]. Farrell et al. have determined that 3,4DMCA is preferentially absorbed in the free form mainly by passive diffusion in the upper gastrointestinal tract [19], whereas dimethoxycinnamic acid derivatives are de-esterified by enteric microflora esterases. Therefore, a significant amount of free 3,4DMCA is transported to the serosal side increasing its total amount in plasma in comparison to its baseline concentration in coffee solution. Under pharmacokinetics studies, it was also confirmed that 3,4DMCA may undergo reduction with colonic microfloral reductase to dimethoxydihydrocinnamic acid, which small amounts have been quantified in plasma of volunteers [19].

Experiments performed with human hepatic S9 fraction as a model to characterize the metabolic stability and human Caco-2 cell monolayer as a well-accepted model for assessing the cellular permeability of active molecules candidates have proven that methylated polyphenols exhibit higher intestinal permeability as well as higher metabolic stability than their corresponding unmethylated forms [20]. For 3,4DMCA it has been confirmed that the presence of two methoxy groups in its structure determines the activity inhibition of enzymes of first-pass metabolism in the liver. Thereby, this compound reaches the bloodstream in unchanged form in the contrast to the hydroxy derivatives of CA, which are transformed in the human body into sulfonic and glucuronic products that significantly lower their biological activity and oral bioavailability [21]. Farrell et al. reported also that cell membrane permeability of 3,4DMCA is almost 10-fold greater than its conjugates with quinic acid [19].

Although methylated phenols are characterized by even 5- to 8-fold higher rate of absorption than the corresponding unmethylated analogues [20], their therapeutic potential and pro-health activity are still limited by their low content in dietary products, low solubility and rapid metabolism. Therefore, in the last two decades, many attempts have been made to change their physicochemical properties in order to broaden the possibilities of their practical application. The process of lipophilization has been extensively studied as the method for enhancement of polyphenols' bioavailability and therapeutic efficacy after oral administration. In the example of mycophenolic acid triglyceride conjugate, Porter's group confirmed that conjugates of phenols with lipid molecules are targeted to the lymphatic system and are transported through [22,23]. It is an additional advantage in the view of the significant role of the lymphatic system in diseases progression, particularly in therapeutic areas such as autoimmune disorders, cancer or metabolic syndrome [24]. Siliphos®, a phosphatidylcholine complex of silybin has been reported to exhibit 5-fold higher bioavailability after oral administration than free silybin [25]. Recently, we have also shown that the lipophilization of methoxy derivatives of benzoic acid and cinnamic acid based on their direct conjugation with glycerol backbone of phosphatidylcholine leads to greatly increased anticancer activity in the comparison to the free form of these acids [26,27].

Due to the fact that phospholipid complexes are rather unstable in human body and chemical synthesis of phenolipids could be difficult, the promising alternative are the biocatalytic processes of their production. Lipase-catalyzed reactions offer a high selectivity and milder reaction conditions as well as a more natural approach as compared to its chemical counterpart [28]. Many reports on the production of esters and structured triacylglycerols (TAGs) containing a phenolic acid moiety are presented in the literature, where lipases play a key role as biocatalysts [29–31]. However, according to our knowledge enzymatic lipophilization of 3,4DMCA was demonstrated only by two research groups. Synthesis of 3,4-dimethoxycinnamic esters by direct esterification of 3,4DMCA with fatty alcohols using Novozym 435 as a biocatalyst have been described by Guyot et al. Based on the results

obtained, the authors indicated that decrease in the esterification yield from 60% to 12% was the result of increasing alcohol carbon chain length from 4 to 8, respectively [32]. In turn, Karboune et al. have shown that it is possible to obtain TAGs structured with 3,4DMCA during the lipase-catalyzed acidolysis of flaxseed oil with this acid. However, the bioconversion efficiency of this process was on a very low level of only 7%, which according to the authors could be the result of spatial hindrance caused by the presence of two methoxy groups in the aromatic ring [33]. Since our previous studies clearly showed that it is possible to obtain in biocatalytic process the phospholipid derivatives of ferulic [34] and anisic [35] acid, we decided to apply this method for lipophilization of 3,4DMCA. For this purpose, we used the egg-yolk phosphatidylcholine (PC) as the lipid substrate which is known to be an effective carrier of substances of hydrophilic nature and is characterized by high compatibility with biological membranes and able to penetrate even the blood–brain barrier [36,37]. In addition, the PC is an easily digestible source of polyunsaturated fatty acid (PUFA) and choline which have a strong health-promoting effect on the body [37,38].

The aim of our research was to develop and optimize a lipase-catalyzed synthesis of structured phospholipids (PLs) containing 3,4-dimethoxycinnamic acid. The activity and selectivity of enzymes and, consequently, the efficiency of enzymatic reaction depends on many parameters such as organic solvent, temperature, substrate molar ratio, reaction time, type and content of biocatalyst. Therefore, it is necessary to evaluate the influence of all of them on the effectiveness of the enzymatic reaction. The traditional method of enzymatic process optimization is based on the study of one factor at a time, which is a time-consuming procedure that incurs higher costs. Nowadays, application of various statistical experimental design to the optimization of enzymatic synthesis is extensively applied. This method is demonstrated to be efficient for understanding the relationship between independent and dependent variables in biocatalytic processes [39,40]. Among many statistical models available, the Box-Behnken design (BBD) is one of those considered to be the most efficient [41]. A significant advantage of BBD is the possibility of construction a second-order polynomial equation and surface plots to predict responses. Therefore, at the beginning of our experiments we performed preliminary tests to determine the initial reaction conditions represented by non-numerical variables (acyl residue donor, biocatalyst and organic solvent), and then we applied Box-Behnken design to optimize numerical variables such as substrate molar ratio, enzyme loading and reaction time.

## 2. Results

### 2.1. Effect of 3,4-Dimethoxycinnamonyl Donors on the Transesterification Reaction

The low solubility of phenols and its methoxy derivatives in lipophilic environment may strongly limit the efficiency of reactions catalyzed by lipases. We have already observed this phenomenon during a study on the enzymatic modification of PC with ferulic acid (FA) [34]. Using FA as an acyl donor we obtained much lower incorporation into phosphatidylcholine than under the experiments performed with its ethyl ester (EF). Therefore, we have started our investigation of the lipophilization process from the comparison of the acidolysis and interesterification reactions using 3,4-dimethoxycinnamic acid (3,4DMCA) and its ethyl ester (E3,4DMCA) as the acyl donors. The initial conditions applied for both reaction systems (acidolysis and interesterification) were proposed on the basis of the literature data and were as follows: 1/10 PC/acyl donor molar ratio, 30% enzyme dosage, temperature 50 °C and heptane as the reaction medium. Novozym 435 in a dose 30% (w/w) was chosen due to its high efficiency demonstrated previously in triacylglycerols modification carried out with 3,4DMCA [33]. Optimal temperature reported in the literature for the transesterification reactions catalyzed by Novozym 435 falls in the range of 40–60 °C, however, in our previous study we proved that 50 °C is the most suitable for PLs modifications [35]. Heptane was used as a highly hydrophobic reaction medium, while substrate molar ratio was set as 1/10 PC/acyl donor.

The progress of the reaction was monitored after 1, 2, 3 and 4 days, respectively. To confirm the degree of incorporation of 3,4DMCA into PC at a selected time intervals, samples were taken

and purified by solid-phase extraction (SPE) to obtain pure PLs fractions which were subsequently derivatized and analyzed by gas chromatography (GC). At this stage of optimization of reaction parameters, phospholipid products were not fractionated. The incorporation degree of 3,4DMCA has been evaluated in the total phospholipid fraction. The screening experiments have shown that there is a significant difference between the degree of incorporation of 3,4DMCA into PLs during acidolysis and interesterification reactions (Figure 1A). It was observed that the formation of *O*-methylated phenophospholipids is generally possible only during the reaction with ethyl ester of 3,4-dimethoxycinnamic acid as an acyl donor. Furthermore, under the proposed conditions the interesterification reaction had a linear trend affording the highest degree of incorporation (21 mol%) after 4 days, while in the case of acidolysis it was only 2 mol%. Based on these, E3,4DMCA was selected as a better acyl donor for further studies on the enzymatic synthesis of 3,4-dimethoxycinnamoylated phospholipids.

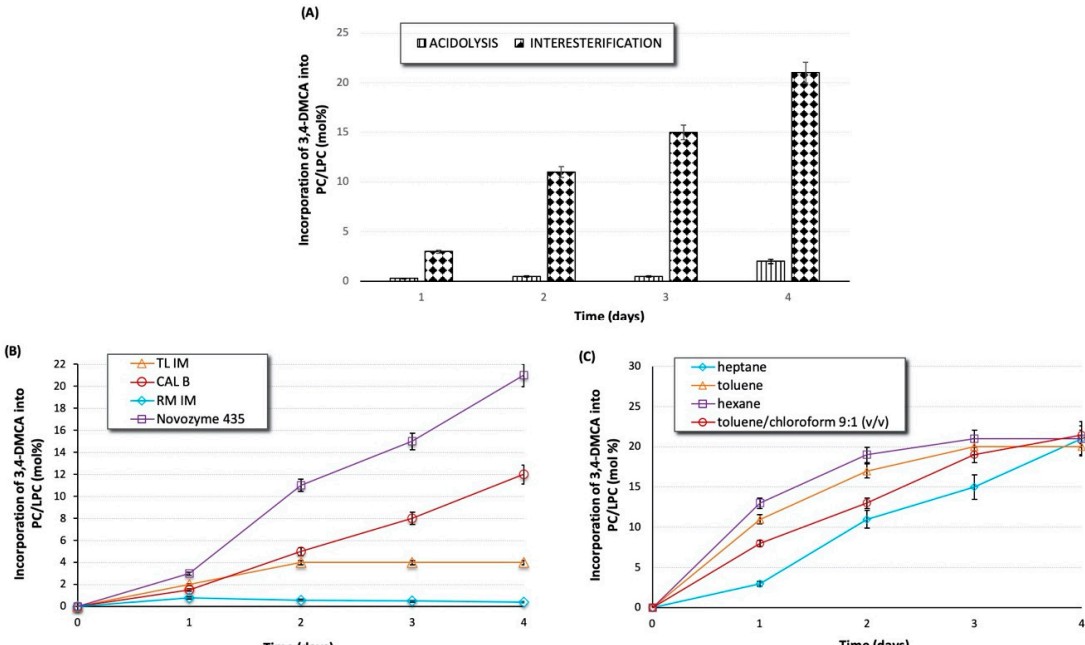

**Figure 1.** (**A**) Effect of different acyl donors on the incorporation of 3,4DMCA into egg-yolk phosphatidylcholine (PC). Reaction conditions: solvent, heptane; PC/acyl donor molar ratio, 1/10; lipase dosage, Novozym 435 30% (w/w); temperature, 50 °C; (**B**) time course of different lipase-catalyzed transesterification between egg-yolk phosphatidylcholine and 3,4-dimethoxycinnamic acid ethyl ester. Reaction conditions: heptane; PC/E3,4DMCA, 1/10; lipase dosage 30% (w/w); 50 °C; (**C**) effect of different organic solvent on the transesterification reaction of PC with E3,4DMCA. Reaction conditions: PC/E3,4DMCA, 1/10; Novozym 435 30% (w/w); 50 °C.

## 2.2. Screening of Lipase on the Interesterification Reaction

The reaction of interesterification catalyzed by lipases is quite a new process developed in the early 1980s by Unilever, Novozymes and Fuji, in which lipases are used to exchange of fatty acids residue in lipid molecules [42]. Lipases are the enzymes that catalyzes the complete or partial hydrolysis of ester bounds in TAGs or PLs and under appropriate conditions can promote ester formation as well. Two main classes of lipases used for interesterification can be distinguished: non-specific and 1,3-specific. For modification of egg-yolk phosphatidylcholine with 3,4DMCA, we selected those lipases which are reported in the literature to be highly selective towards the *sn*-1 position of TAGs and PC [29,33,34,43,44], keeping in mind natural therapeutic potential of PC resulting from presence of unsaturated fatty acids in its *sn*-2 position.

In enzymatic processes, lipases can be used in a free powdered form or immobilized on diverse carriers. These second form has many advantages and in positive way influence on their recoverability, reusability and stability, making the interesterification process economically more feasible. In our study we screened four commercially available immobilized lipases: Lipozyme®, Lipozyme TL IM, CALB and Novozym 435 by incubating them in a dosage of 30% (w/w) with 1/10 mole ratio of PC and 3,4DMCA ethyl ester at 50 °C in heptane. Figure 1B shows the effect of different enzymes on 3,4DMCA incorporation into the PLs fraction. Three among the tested enzymes: Novozym 435, CALB and TL IM, were characterized by the ability to incorporate 3,4-DMCA into the PC whereas RM IM was shown not to be active. The incorporation degree of 3,4DMCA attained with various lipases was in order of Novozym 435 > CALB > Lipozyme TL IM. However, it should be noted that only for lipases B from *Candida antarctica* immobilized on different carriers we observed significant activity in this area. In the reaction catalyzed by Novozym 435 the incorporation reached a maximum 21 mol% within 4 days, whereas in CALB-catalyzed interesterification 12 mol% of incorporation was observed. In this investigation, Lipozyme TL IM showed much lower activity giving only 4 mol% incorporation of 3,4DMCA into the PLs fraction. In our previous studies, Novozym 435 was also the most efficient in incorporation of citronellic acid, ferulic acid and anisic acid into phosphatidylcholine in the acidolysis and interesterification processes [34,35,44]. Therefore, this biocatalyst was also used further in this work.

### 2.3. Effect of Solvent on the Interesterification Reactions

The effectiveness of interesterification reactions can be enhanced by using appropriate solvent as a medium which reduces the reaction viscosity. Then it is also possible to shift equilibrium of lipase-catalyzed reactions towards transesterification products because appropriate organic solvent can limit the water activity in the environment. However, care should be taken to ensure that the solvent is not toxic to the enzyme and dissolves the substrates as well. Based on that, three organic solvents, *n*-hexane (log $P$= 3.5), *n*-heptane (log $P$ = 4) and toluene (log $P$ = 2.5), were selected in the studies. Previously it has been reported that usage of a binary solvent system improves the solubility of ferulic acids in modification of phospholipids [45], therefore we also evaluated the described mixture of organic solvents toluene/chloroform in a volume ratio of 9:1 (log $P$ = 2.5/2.01).

As shown in Figure 1C, in all tested solvents the process of interesterification proceeded effectively giving the degree of incorporation of 3,4-DMCA into the PLs fraction ranging from 19 to 21 mol%. The slowest rate of process was observed in the experiment carried out in *n*-heptane. In this solvent incorporation of dimethoxycinnamic acid reached 21 mol% after 4 days whereas in *n*-hexane the same incorporation level was achieved after 3 days of reaction.

### 2.4. Statistical Analysis of Enzymatic Interesterification of Phosphatidylcholine (PC) with Ethyl Ester of 3,4-Dimethoxycinnamic Acid (E3,4DMCA)

In order to reduce the costs of experiments and, at the same time, better understand the impact of independent variables on the degree of incorporation of 3,4DMCA into phosphatidylcholine and correlations between them after the preliminary selection of biocatalyst and reaction medium, response surface methodology (RSM) and Box-Behnken design were employed. We prepared a 3-level, 3-factor statistical design to evaluate the effects of the selected variables, i.e., substrate molar ratio PC/E3,4DMCA, enzyme loading and reaction time on the synthesis of 3,4-dimethoxycinnamoylated phospholipids (3,4DMCA-PLs). Table 1 shows experimental and predicted values of incorporation expressed in mol%, which were obtained as a result of 15 number of runs with 3 replicates at the central point. The predicted values were obtained from a model fitting technique using the software Statistica 13.3 (StatSoft, Inc. (Tulsa, OK, USA)) and were seen to be sufficiently correlated to the experimental values (Table 1).

**Table 1.** Experimental and predicted value of incorporation of 3,4-dimethoxycinnamic acid (3,4DMCA) into phosphatidylcholine/lysophosphatidylcholine (PC/LPC) fraction in the Box-Behnken design.

| Run | ($X_1$) Substrate Molar Ratio PC/E3,4DMCA | ($X_2$) Enzyme Load [%] | ($X_3$) Reaction Time [days] | Incorporation of 3,4DMCA into PC/LPC [mol%] [a] (Experimental) | Incorporation of 3,4DMCA to PC/LPC [mol%] (Predicted) |
|---|---|---|---|---|---|
| 1 | 5 | 20 | 3 | 2 ± 0.3 | 4 |
| 2 | 15 | 20 | 3 | 15 ± 0.6 | 14 |
| 3 | 5 | 40 | 3 | 7 ± 0.4 | 8 |
| 4 | 15 | 40 | 3 | 20 ± 0.8 | 18 |
| 5 | 5 | 30 | 2 | 5 ± 0.7 | 5 |
| 6 | 15 | 30 | 2 | 10 ± 0.6 | 13 |
| 7 | 5 | 30 | 4 | 6 ± 0.9 | 3 |
| 8 | 15 | 30 | 4 | 13 ± 0.6 | 13 |
| 9 | 10 | 20 | 2 | 19 ± 0.1 | 17 |
| 10 | 10 | 40 | 2 | 19 ± 0.1 | 18 |
| 11 | 10 | 20 | 4 | 12 ± 0.4 | 13 |
| 12 | 10 | 40 | 4 | 18 ± 0.3 | 20 |
| 13 | 10 | 30 | 3 | 21 ± 0.5 | 21 |
| 14 | 10 | 30 | 3 | 21 ± 0.7 | 21 |
| 15 | 10 | 30 | 3 | 21 ± 0.2 | 21 |

[a] Data are presented as mean ± standard deviation (SD) of two independent analyses.

Analysis of variance (ANOVA) was performed for each of the 3 evaluated variables (Table 2). The model of F-value was identified as significant as the *p*-value was less than 0.05. A high value of coefficient of determination value ($R^2$) 0.93214 (close to 1) indicates that polynomial equation model was highly adequate to represent the actual relationship between the response and variables. Normally, a regression model with the $R^2$ value above 0.9 is considered as a model having high correlation [46]. Furthermore, the linear distribution visible in Figure 2A, which compares the predicted and experimental values of degree of incorporation, also confirms good fit of the model. The generated model was employed subsequently to study the effect of various parameters and their interactions ($X_1$ by $X_2$, $X_1$ by $X_3$ and $X_2$ by $X_3$) on the incorporation degree. Based on ANOVA results it is visible that from 3 tested variables only substrate molar ratio (Q and L) has a statistically significant effect ($p < 0.05$) on incorporation while the other two parameters: enzyme load (Q, L) and reaction time (Q, L) and the interaction between them are not statistically significant ($p > 0.05$). These data are also visible on the Pareto chart in Figure 2B, which additionally illustrates the direction of impact of individual effects. Most of the independent variables included in this chart have positive value, which means that with an increase in their value, the degree of incorporation also increases. Negative values of a coefficient estimate denote a negative influence of parameters on the incorporation degree.

**Table 2.** Analysis of variance (ANOVA) for interesterification variables pertaining to the response of percent incorporation of 3,4DMCA to PC/LPC.

| Evaluated Factors | Sum of Squares | Degrees of Freedom | Medium Square | F-Value | *p*-Value |
|---|---|---|---|---|---|
| ($X_1$) Substrate molar ratio (L) | 180.5000 | 1 | 180.5000 | 24.33708 | 0.002622 |
| Substrate molar ratio (Q) | 342.2500 | 1 | 342.2500 | 46.14607 | 0.000498 |
| ($X_2$) Enzyme load (L) | 32.0000 | 1 | 32.0000 | 4.31461 | 0.083060 |
| Enzyme load (Q) | 2.2500 | 1 | 2.2500 | 0.30337 | 0.601667 |
| ($X_3$) Time of reaction (L) | 2.0000 | 1 | 2.0000 | 0.26966 | 0.622149 |
| Time of reaction (Q) | 42.2500 | 1 | 42.2500 | 5.69663 | 0.054265 |
| $X_1$ by $X_2$ | 0.0000 | 1 | 0.0000 | 0.00000 | 1.000000 |
| $X_1$ by $X_3$ | 1.0000 | 1 | 1.0000 | 0.13483 | 0.726077 |
| $X_2$ by $X_3$ | 9.0000 | 1 | 9.0000 | 1.21348 | 0.312860 |
| Error | 44.5000 | 6 | 7.4167 | | |
| Total error | 655.7500 | 15 | | | |
| $R^2 = 0.93214$ | | | | | |

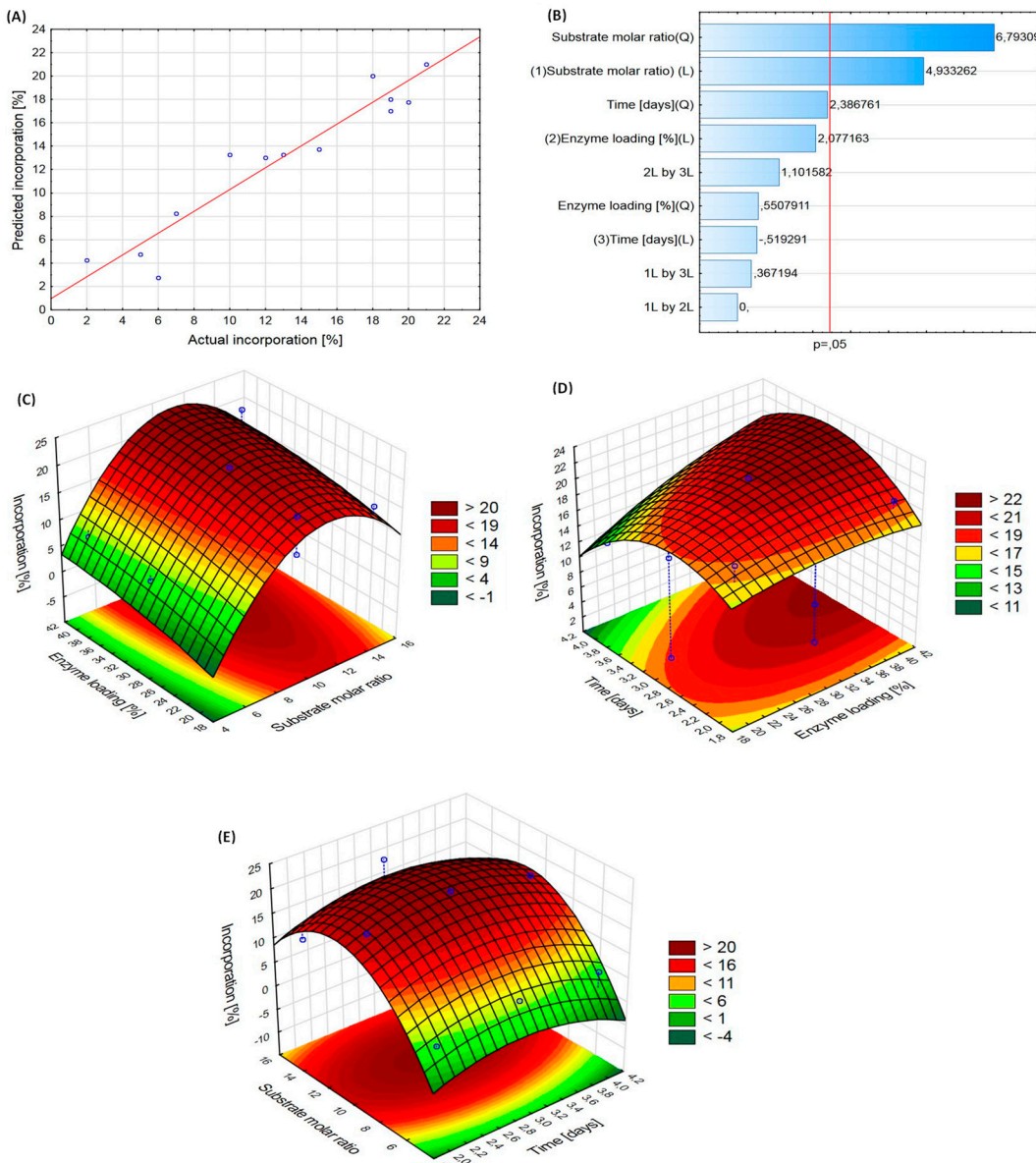

**Figure 2.** (**A**) Correlation of actual and predicted values of incorporation of 3,4DMCA into phospholipids (PLs) by the response surface model, (**B**) Pareto chart of the analyzed effects for incorporation of 3,4DMCA into PLs, Response surface plot showing effect of (**C**) enzyme loading and substrate molar ratio (**D**) time of reaction and enzyme loading and (**E**) substrate molar ratio and time of reaction on the incorporation of 3,4DMCA into PLs.

To better understand the relationship between reaction parameters and responses, the three-dimensional (3D) response surface plots generated from the model were applied (Figure 2C–E). In Figure 2C the effect of varying of substrate molar ratio and enzyme dosage on incorporation of 3,4DMCA into PLs fraction was studied when time of reaction was kept constant (3 days). The results show that an increase of substrate molar ratio in the range from 1:8 to 1:12 results in increasing level of incorporation from 14 to 21 mol% and the dosage of enzyme does not have significant impact on the reaction. These results are in agreement with the literature data. Yang et al. reported that a high molar ratio of the acyl donor to the lipid shifts the reaction balance towards products and improves the process of incorporation of acyl residues [47]. However, it should be noted that in the case of synthesis of 3,4-dimethoxycinnamoylated phospholipids, a further increase in the PC:E3,4DMCA ratio to 1:16 causes a decrease in the degree of incorporation to almost 10 mol%, which may be the result of the

inhibitory effect of too high concentration of substrate on the enzyme activity. Lack of a relationship between higher enzyme dose and reaction productivity have been also observed before by Chen who reported that when the content of the enzyme in the reaction mixture reaches a certain value, further increasing its dose does not cause further increase in the productivity of modified phospholipids [45,48]. Similar observations can be made from Figure 2D, where constant parameter was substrate molar ratio 1:10. It is visible that to short (less than 2 days) or to long (more than 3 days) reaction time causes rapid reduction of the degree of incorporation from the highest value 21 mol% to 10 mol% while the change in the enzyme content in the tested range is of little significance. However, if we consider the relation between varying reaction time and substrate molar ratio (enzyme dosage is constant) depicted in Figure 2E it is visible that in this case reaction time has less impact on the synthesis of structured PC than substrate molar ratio. This observation is in agreement with the conclusions drawn on the basis of ANOVA analysis, which indicates that the substrate molar ratio has the greatest impact on the course of the phospholipid modification process (Figure 2B).

## 2.5. Identification of the Reaction Products

The optimal parameters for Novozym 435-catalyzed synthesis of 3,4-dimethoxycinnamoylated phospholipids were determined using response surface methodology. The selected optimized parameters were established as PC/E3,4-DMCA molar ratio 1/10, enzyme load 30% (w/w), reaction medium hexane, temperature 50 °C and incubation time 3 days. At these optimal reaction conditions, the interesterification of egg-yolk phosphatidylcholine with ethyl ester of 3,4-dimethoxycinnamic acid was conducted in larger scale. Below we present the reaction scheme of enzymatic interesterification of egg-yolk PC with ethyl ester of 3,4-dimethoxycinnamic acid catalyzed by Novozym 435 (Figure 3).

**Figure 3.** Reaction scheme of lipase-catalyzed interesterification of egg-yolk PC and E3,4DMCA.

The products were identified by thin-layer chromatography (TLC), gas chromatography (GC), high-performance liquid chromatography (HPLC) and nuclear magnetic resonance (NMR) analysis. The products obtained were first qualitatively identified by TLC. Based on the comparison of the $R_f$ values of standards: PC-egg ($R_f$ = 0.45), LPC-egg ($R_f$ = 0.1), 3,4-DMCA ($R_f$ = 0.85) and E3,4-DMCA ($R_f$ = 0.72) with products of the reaction mixtures spotted on the TLC plates we identified two new bands slightly more polar than natural PC and LPC with $R_f$ value 0.35 and 0.07, respectively.

The modified LPC as a more hydrophilic product was analyzed by reversed phase HPLC using a UV/DAD detector under UV light 310 nm in which the aromatic ring attached to the glycerol backbone of LPC produces fluorescent absorption ($R_t$ = 4.991). Formation of the minor more lipophilic product 3,4DMCA-PC was monitored by normal phase HPLC equipped with a UV/CAD detector (at 310 nm) ($R_t$ = 11.433). The HPLC chromatograms of the substrates and products are shown in Figures S5 and S6 of Supplementary Materials.

In the next step, new products were separated from fatty acids and unreacted ester of 3,4-dimethoxycinnamic acid as well as form PC-egg and LPC-egg and fractioned by the column chromatography. The main product 3,4-dimethoxycinnamoylated-LPC (3,4DMCA-LPC) was obtained in high 27.5% of isolated yield whereas second product 3,4-dimethoxycinnamoylated-PC (3,4DMCA-PC) was obtained in trace amount only in 3.5% isolated yield.

The structure of purified product 3,4DMCA-LPC was fully confirmed by NMR spectral data (Supplementary Materials Figure S1). From $^{1}$H NMR spectral data of modified LPC, a singlet at δ 2.92 indicates the presence of 9 protons adjacent to nitrogen of the choline moiety. The multiplicity range in δ 6.07–7.36 indicates the aromatic and double bond protons of the aromatic acid. From $^{13}$C NMR spectral data of this product, the signal observed at 167.69 ppm indicates the carbonyl group and the signals in the range 111.20–149.21 ppm indicate the aromatic and double bond carbons (Supplementary Materials Figure S2).

The fatty acids composition was also evaluated in the modified phospholipid fraction and product. Table 3 shows the comparison of the fatty acid profile in modified phospholipid fractions and obtained 3,4DMCA-PC. It is visible that incorporation of 3,4DMCA into phospholipid fraction obtained as a result of interesterification reached 21 mol%, whereas GC analysis of the products after their isolation and purification showed, that the content of 3,4DMCA in modified PC (3,4DMCA-PC) was only 8 mol%. It is also noticeable that with the increase of incorporation of phenolic acid, the content of saturated fatty acids (present mainly in the *sn*-1 position of natural phospholipids) decreases, what confirms the regioselectivity of Novozym 435.

**Table 3.** Fatty acid composition (% according to gas chromatography (GC)) of phospholipid fractions obtained of interesterification reaction of egg-yolk PC with 3,4-DMCAE.

| Fatty and 3,4-DMCA Acids | Native PC [mol%] [a] | Modified Phospholipid Fraction PC/LPC [mol%] [a] | 3,4-DMCA-PC [mol%] [a] |
|---|---|---|---|
| C16:0 (PA) | 34 ± 0.2 | 11 ± 0.4 | 11 ± 0.9 |
| C16:1 (OPA) | 1 ± 0.7 | 1 ± 0.2 | 2 ± 0.1 |
| C18:0 (SA) | 15 ± 0.9 | 6 ± 0.7 | 8 ± 0.4 |
| C18:1 (OA) | 26 ± 0.7 | 30 ± 0.7 | 43 ± 0.3 |
| C18:2 (LA) | 20 ± 0.3 | 25 | 26 ± 0.2 |
| C20:4 (AA) | 4 ± 0.2 | 6 ± 0.3 | 2 ± 0.2 |
| 3,4-DMCA | - | 21 | 8 ± 0.2 |

[a] Data are presented as mean ± SD of two independent analysis.

## 3. Materials and Methods

### 3.1. Substrates, Chemicals and Enzymes for Enzymatic Reactions

We synthesized 3,4-dimethoxycinnamic acid ethyl ester (E3,4DMCA) in high 86% yield according to the method described previously [49]. Its purity 98% was confirmed by GC whereas spectroscopic data were compared with the literature [50]. Native phosphatidylcholine (PC) was isolated from egg-yolk of Lohman Brown hens and purified as described in an earlier paper [44]. The purity of obtained PC was analyzed by TLC on silica gel-coated plates and further confirmed via the HPLC [51].

Lipases from *Candida antarctica* (Novozym® 435, immobilized, >5000 U/g; CALB, immobilized, >1800 U/g) were purchased from Sigma-Aldrich (St. Louis, MO, USA). Lipase from *Rhizomucor miehei* (Lipozyme® RM IM, immobilized, >30 U/g) was provided by Fluka (Buchs, Switzerland) and lipase from *Thermomyces lanuginosus* (Lipozyme® TL IM, immobilized, 250 U/g) was obtained from the Novozymes A/S (Bagsvaerd, Denmark). A boron trifluoride methanol complex solution (13–15% BF$_3$ × MeOH) and sodium methylate were purchased from Sigma-Aldrich (St. Louis, MO, USA). All organic solvents used in chromatography, silica gel-coated aluminium plates (Kieselgel 6- F254, 0.2 mm) used in thin layer chromatography (TLC) and the silica gel (Kieselgel 60, 230–400 mesh) used in the column chromatography, were purchased from Merck (Darmstadt, Germany).

### 3.2. Lipase-Catalyzed Reactions of Acidolysis/Interesterification of PC with 3,4DMCA/E3,4DMCA

Experiments have been started with the evaluation of two methods of enzymatic modification of egg-yolk phosphatidylcholine (PC) acidolysis and interesterification with 3,4DMCA and E3,4DMCA, respectively. Lipase-catalyzed reactions of PC (20 mg, 0.026 mmol) with 3,4DMCA/E3,4DMCA (at molar ratio PC/acyl donor, 1/10) were carried out in 2 mL of heptane in 5 mL screw-capped vials

on a magnetic stirrer (300 rpm) in $N_2$ atmosphere at 50 °C using Novozym 435 (30% by weight of substrates) as a biocatalyst. In the next step of the study, the type of lipases (Novozym 435, CALB, RM IM, TL IM) and organic solvent (heptane, hexane, toluene, toluene:chloroform 9:1 (*v/v*)) were also tested for the interesterification reaction in another set of experiments. In all experiments after selected time intervals (1, 2, 3 and 4 days) the reactions were stopped by enzyme filtration (G4 Shott funnel with Celite layer). Phospholipid fractions were purified from free fatty acid and unreacted ester by the SPE methodology using silica gel columns (Discovery® DSC-Si SPE, 52654-U 500 mg) according to the procedure described by Rychlicka [34]. The purified phospholipid fraction was next analyzed by thin-layer chromatography (TLC) (3.5.1) and then their acid profile was analyzed by gas chromatography (GC) (3.5.2). The incorporation of 3,4DMCA into phospholipid fraction was expressed as mol% and its quantitative analysis was performed on the basis of the peak areas using GC Chemstation Version A.10.02.

### 3.3. Experimental Factorial Design

After finishing the preliminary tests, a 15-run, 3-factor, 3-level Box-Behnken design was employed to construct polynomial models for the optimization of the process enzymatic interesterification egg-yolk phosphatidylcholine with E3,4DMCA. The independent variables: substrate molar ratio (1:5–1:15), enzyme loading (20–40%), reaction time (2–4 days) were analyzed (Table 1) The STATISTICA 13.3 (StatSoft, Inc.) was used to determine the regression and graphical analysis of the results obtained. Predicted value of dependent variable (% incorporation of 3,4DMCA into phospholipid fraction PC/LPC) was calculated according to the polynomial equation as follows:

$$Y_i = \beta_0 + \beta_{1 \times 1} + \beta_2 X_2 + \beta_3 X_3 + \beta_{12} X_1 X_2 + \beta_{13} X_1 X_3 + \beta_{23} X_2 X_3 + \beta_{11} X_1^2 + \beta_{33} X_3^2$$

where $Y_i$ is predicted response, $\beta_0$ is model constant, $X_1$–$X_3$ are independent variables and $\beta_1$–$\beta_{33}$ are regression coefficient. All experiments were carried out in two independent analyses and the averages of incorporation were taken as the response.

### 3.4. Preparative Scale of Lipase-Catalyzed Interesterification of PC with E3,4DMCA

Reaction of egg-yolk phosphatidylcholine (200 mg, 0.26 mmol) with E3,4DMCA (at molar ratio PC/E3,4DMCA, 1/10) was catalyzed by Novozym 435 (30% by weight of substrates) and carried out in 20 mL of hexane on a magnetic stirrer (300 rpm) in $N_2$ atmosphere at 50 °C. After 3 days interesterification reaction was stopped by enzyme filtration on G4 Shott funnel with Celite layer and organic solvent was evaporated in vacuo. Crude mixture of products was then dissolved in $CHCl_3$ and purified by column chromatography according to the procedure described before [35]. Individual phospholipid fractions PC-egg, LPC-egg, modified phosphatidylcholine (3,4DMCA-PC) and modified lysophosphatidylcholine (3,4DMCA-LPC) were analyzed by TLC plates (3.5.1) GC, HPLC and NMR spectroscopy ([1]H, [13]C, [31]P).

### 3.5. Analytical Methods

#### 3.5.1. Thin-Layer Chromatography (TLC)

Acidolysis or interesterification reaction progress and qualitative analysis of reaction products were controlled by thin-layer chromatography (TLC). For this purpose, samples were spotted on TLC plates and eluted with the mixture of chloroform/methanol/water (65:25:4, *v/v*/v). The products were identify by spraying the TLC plates with the 0.05% primuline solution (acetone: water, 8:2, *v/v*) and then exposing the plates to UV light ($\lambda = 365$ nm).

### 3.5.2. Gas Chromatography (GC)

Analysis of acid profile of reactions products after SPE or column chromatography purification (PC-egg, LPC-egg, 3,4DMCA-PC, 3,4DMCA-LPC) and standards (PC-egg, 3,4DMCA) were analyzed by gas chromatography after their derivatization to the methyl esters according to the procedure described before [34]. GC analysis were conducted on an Agilent 6890N instrument equipped with DB-WAX column (30 m × 0.32 mm × 0.25 μm) manufactured by Agilent (Santa Clara, CA, USA) using the program and settings previously described [34]. To confirm the fatty acid methyl esters (FAME) profile, their retention times were compared with those of a standard FAME mixture (Supelco 37 FAME Mix) purchased from Sigma Aldrich.

### 3.5.3. High-Performance Liquid Chromatography (HPLC)

Reactions product 3,4DMCA-PC was also analyzed by HPLC on an DIONEX UltiMate 3000 chromatograph from Thermo Fisher Scientific (Olten, Switzerland) equipped with UV/CAD detector (at 310 nm). A BetaSil DIOL column (Thermo Scientific, 150 × 4.6 mm, 5 μm) was used for analysis. The injection volume, autosampler and column temperature for all analysis were as follows: 15 μL, 20 and 30 °C. Analysis were performed in a gradient mode with a constant flow of 1.5 mL/min. and started with solvent A (1% HCOOH, 0.1% TEA in water) next solvent B (hexane) and solvent C (2-propanol). The elution program was as follows: 3/40/57 (%A/%B/%C (*v/v/*v)), at 5 min = 10/40/50, at 9 min = 10/40/50, at 9.1 min = 3/40/57 and at 19 min = 43/40/57. Total analysis time was 19 min.

To analyze more hydrophilic product 3,4DMCA-LPC RP-HPLC with Ascentis® Express C18 column (150 × 3.0 mm, 5 μm) and UV/DAD detector (310 nm) was used according to the method described before [34]. Analysis was performed in a binary system of solvents: A (water with 3% of acetic acid) and B (acetonitrile) at a flow rate 0.5 mL/min. The gradient mode was as follows: 0 min: 90% A and 10% B; 10–20 min: maintained 30% A and 70% B; 20–21 min changed to 90% A and 10% B. Total analysis time was 31 min. Samples were diluted in a solvent A and the injection volume was 10 μL.

### 3.5.4. Spectroscopic Spectra (Nuclear Magnetic Resonance, NMR)

NMR analysis was performed to confirm the structure of the products. However, due to the low efficiency of modified PC (3,4DMCA) synthesis NMR experiments was performed only for modified LPC fraction (3,4DMCA-LPC). For this purpose, 3,4DMCA-LPC purified by column chromatography was dissolved in 0.6 mL of CDCl$_3$/MeOH (2:1, *v/v*) in NMR tube and then analyzed on Bruker Advance II 600 MHz spectrometer (Bruker, Billerica, MA, USA).

*1-(3,4-dimethoxy)cinnamoyl-2-hydroxy-sn-glycero-3-phosphocholine (3,4DMCA-LPC)*

Colourless greasy solid (26% yield, R$_f$ 0.75); $^1$H NMR (600 MHz, CDCl$_3$/CD$_3$OD 2:1 (*v/v*)), δ: 2.92 (s, 9H, –N(CH$_3$)$_3$), 3.31 (m, 2H, CH$_2$-β), 3.60 (2s, 6H, –OCH$_3$), 3.63–3.67 (m, 2H, CH$_2$-3'), 3.75 (m, 1H, H-2'), 3.87–3.92 (m, 5H, CH$_2$-1',CH$_2$-α, –OH), 6.07 (d, 1H, *J* = 15.9 Hz, H-2), 6.58 (m, 1H, H-2''), 6.84 (m, 2H, H-5'', H-6''), 7.36 (d, 1H, *J* = 15.9 Hz, H-3); $^{13}$C NMR (150 MHz, CDCl$_3$/CD$_3$OD 2:1 (*v/v*)) δ: 53.84 ((–N(CH$_3$)$_3$), 55.66 (–OCH$_3$), 59.24 (C-α), 62.13 (C-1'), 64.98 (C-β), 66.22 (C-3'), 68.18 (C-2'), 111.20 (Ar), 114.83 (C-2), 122.89 (Ar), 125.69 (Ar), 128.70 (Ar), 131.11 (Ar), 145.68 (C-3), 149.21 (Ar), 167.70 (C-1); $^{31}$P NMR (243 MHz, CDCl$_3$/CD$_3$OD 2:1 (*v/v*)) δ: −2.90.

## 4. Conclusions

The study showed that the commercial immobilized form of lipase B from *Candida antarctica* (Novozym 435) is a good biocatalyst for the synthesis of 3,4-dimethoxycinnamoylated lysophosphatidylcholine (3,4DMCA-LPC) in the interesterification of egg-yolk phosphatidylcholine with ethyl ester of 3,4DMCA in an organic medium. The optimization of this process was performed by statistical design methods and for this purpose Box-Behnken design and the next surface response methodology (RSM) were successfully applied. The $R^2$ value of 0.93214 indicated a good fit of the

model with experimental findings. The ANOVA implied that the model satisfactorily represented the real relationship of the three main reaction variables and the response. The molar ratio showed a significant effect on the productivity of the reaction. The highest incorporation of 3,4DMCA into phospholipid fraction was achieved at 1/10 PC/E3,4DMCA. We obtained 3,4-dimethoxycinnamoylated lysophosphatidylcholine (3,4DMCA-LPC) in a high isolated yield of 27.5% (w/w), whereas modified phosphatidylcholine (3,4DMCA-PC) was formed during the reaction of interesterification as the minor product. We obtained 3,4-dimethoxycinnamoylated phosphatidylcholine (3,4DMCA-PC) in trace amount in only 3.5% of isolated yield.

**Supplementary Materials:** The following are available online at http://www.mdpi.com/2073-4344/10/5/588/s1, Figure S1: $^1$H NMR spectrum of 3,4DMCA-LPC, Figure S2: $^{13}$C NMR spectrum of 3,4DMCA-LPC, Figure S3: $^{31}$P NMR spectrum of 3,4DMCA-LPC, Figure S4: HPLC chromatogram of 3,4DMCA-PC, Figure S5: HPLC chromatogram of 3,4DMCA-LPC, Figure S6: GC chromatogram of fatty acid composition of PC-egg (standard), Figure S7: GC chromatogram of methyl ester of 3,4DMCA (standard), Figure S8: GC chromatogram of fatty acid composition of modified phospholipid fraction PC/LPC, Figure S9: GC chromatogram of fatty acid composition of isolated product of enzymatic reaction of 3,4DMCA-PC.

**Author Contributions:** Conceptualization, A.G.; Investigation, M.R.; Methodology, M.R.; A.G.; Visualization M.R.; A.G.; Validation, A.G.; N.N.; Writing—original draft preparation, M.R. and A.G.; Writing—Review and editing, A.G; M.R; N.N.; All authors have read and agreed to the published version of the manuscript.

**Funding:** Article Processing Charge (APC) was financed under the Leading Research Groups support project from the subsidy increased for the period 2020–2025 in the amount of 2% of the subsidy referred to Art. 387 (3) of the Law of 20 July 2018 on Higher Education and Science, obtained in 2019.

**Conflicts of Interest:** The authors declare no conflict of interest.

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
