# Peer review of "Development and Optimization of Lipase-Catalyzed Synthesis of Phospholipids Containing 3,4-Dimethoxycinnamic Acid by Response Surface Methodology"

_catalysts, doi:10.3390/catal10050588_

Round 1
Reviewer 1 Report
The article entitled "Development and Optimization of Lipase-Catalyzed Synthesis of Phospholipids Containing 3,4-dimethoxycinnamic Acid by Response Surface Methodology" sent to Catalysts, concerns the current and interesting from the point of view of catalysis topics. Authors carried out enzymatic interesterification using a Novozym 435 biocatalyst, which was selected after previous tests for 4 catalytic systems. The optimum parameters for Novozym 435-catalyzed synthesis of 3,4-dimethoxycinnamoylated phospholipids were determined using response surface methodology.The yield of the enzymatic reaction for the main product of about 27% is acceptable as for biocatalytic reactions. One may wonder, however, whether about 3% of the by-product allows a clear statement on the selectivity of the reaction.
The article is written clearly and comprehensibly. The experimental part is correctly described and the results are well interpreted. A good solution would be to place the chromatograms in the main text - not in supplementary materials, but I leave it to the authors decision.In Figure 3, the structures could be larger and thickened so that the mechanism was clear and visible.
I think that after minor editorial corrections, the article will be ready for publication in Catalysts.
Author Response
May 16, 2020
Anna Gliszczyńska
Department of Chemistry,
Wroclaw University of Environmental and Life Sciences,
Norwida 25, PL-50-375 Wroclaw, Poland
E-mail: anna.gliszczynska@wp.pl
Editor
Catalysts
Dear Editor,
We would like to thank you very much for giving us the possibility to improve the manuscript of our paper “Development and Optimization of Lipase-Catalyzed Synthesis of Phospholipids Containing 3,4-dimethoxycinnamic Acid by Response Surface Methodology” (Catalysts-804932). Here we would like to present and comment on the changes we have made according to the Reviewers’ suggestions and give some explanations of some issues that had been raised by Reviewers.
General comments:
All changes were highlighted in yellow in revised version of the manuscript.
According to Reviewers comments the manuscript was also corrected by a native speaker.
According to Editor comments:
I agree with the reviewer's comments, please revise based on the reviewer’s comments. Moreover, I have some comments.
1. There are no water in this reaction system, so how did hydrolysis occur in the first step in Fig.3. A hydrolysis reaction needs water. Please explain clearly or If you are not sure, please used a simple scheme as ref 44. A simple scheme of interesterification can be found here, https://www.intechopen.com/books/phenolic-compounds-natural-sources-importance-and-applications/synthesis-and-characterization-of-phenolic-lipids.
2. The calculation of Incorporation should be defined in Materials and Methods section 3.2.
Response:
According to the Editor suggestions we have simplified the scheme of interesterification reaction. We defined the calculation of incorporation in section 3.2.
According to the Reviewer 1 comments:
One may wonder, however, whether about 3% of the by-product allows a clear statement on the selectivity of the reaction.
Response:
Novozym 435 is considered as a non-regiospecific lipase however in the majority of lipid modifications showed higher selectivity towards the sn-1 position of TAGs and PC. Formation of trace amount of the by-product seems to confirm higher selectivity of this lipase towards the sn-1 position what is also visible in the decrease of the content of saturated fatty acids in the fatty acids composition of modified phospholipid fraction.
The article is written clearly and comprehensibly. The experimental part is correctly described and the results are well interpreted. A good solution would be to place the chromatograms in the main text - not in supplementary materials, but I leave it to the authors decision. In Figure 3, the structures could be larger and thickened so that the mechanism was clear and visible.
I think that after minor editorial corrections, the article will be ready for publication in Catalysts.
Response:
We decided to leave the chromatograms and spectra in the Supplementary Materials and do not extend the manuscript. We have simplified the scheme of interesterification reaction making them clearer and present the structures in larger scale with higher resolution.
According to the Reviewer 2 comments:
The standard of English needs addressing throughout, as illustrated by several points made regarding the abstract (1-6) and other sections (7-..) below, but the errors in the manuscript are too numerous and frequent to list in a refereeing process.
- Page 1 Abstract Line 13 ‘transesterification’? not sure what ‘interesterification’ is
Transesterification is a one-step process of modification which involves lipid molecule (PC) and acyl donors. Depending on the type of use acyl donor tow reactions can be distinguished acidolysis when acids are used or interesterification when ester is used as an acyl donor. Therefore, in the abstract we used the term of interesterification.
- Page 1 Abstract Line 13 – ‘the ethyl ester’ and check throughout for use of definite article
- Line 15 ‘reaction medium’
- Line 17 ‘have on the process’
- Line 20-21 – what is meant by ‘substrate molar ratio’ – molar equivalency?
- Line 19 ‘for the maximization of’
- Line 38 ‘cinnamic’
- Line 39 what are ‘berries’ – specify
- Line 55 ‘It causes that’ rephrase
10, Line 130 ‘may strongly limit’
We corrected all errors.
11.Line 162 – In the pdf suppled the text in Figure 1 graphs is not legible, especially in the boxes
We correct the Figure 1.
- Line 190-191 ‘were shown not to be active’
- Table 1 -need to harmonise use of sig figs in Table
- Line 303 ‘qualitatively’
We corrected all errors and harmonized data in Table.
- Line 319 – have the NMRs for this product been supplied? The data are there but would be good to include the spectrum as a Figure
We decided to leave the chromatograms and spectra in the Supplementary Materials and do not extend the manuscript. We put in the text the information about the available chromatograms and spectra in Supplementary Materials.
- Experimental -use s, min, d, h, throughout
We corrected all errors.
We would like to express our thanks to Reviewers for very valuable comments, which help us to improve our manuscript. We hope that our explanations and corrections are sufficient, and our paper will be finally accepted for publication.
With best regards,
Reviewer 2 Report
The authors submit a manuscript describing the esterification of phosphatidylcholine using CAL-B and a cinnamate ester as the acyl donor. It builds on previous work by the group in the functionalisation of PC using phenols.
The standard of English needs addressing throughout, as illustrated by several points made regarding the abstract (1-6) and other sections (7-..) below, but the errors in the manuscript are too numerous and frequent to list in a refereeing process.
- Page 1 Abstract Line 13 ‘transesterification’? not sure what ‘interesterification’ is
- Page 1 Abstract Line 13 – ‘the ethyl ester’ and check throughout for use of definite article
- Line 15 ‘reaction medium’
- Line 17 ‘have on the process’
- Line 20-21 – what is meant by ‘substrate molar ratio’ – molar equivalency?
- Line 19 ‘for the maximization of’
- Line 38 ‘cinnamic’
- Line 39 what are ‘berries’ – specify
- Line 55 ‘It causes that’ rephrase
10, Line 130 ‘may strongly limit’
11.Line 162 – In the pdf suppled the text in Figure 1 graphs is not legible, especially in the boxes
- Line 190-191 ‘were shown not to be active’
- Table 1 -need to harmonise use of sig figs in Table
- Line 303 ‘qualitatively’
- Line 319 – have the NMRs for this product been supplied? The data are there but would be good to include the spectrum as a Figure
- Experimental -use s, min, d, h, throughout
Author Response

(The authors gave the same response as above.)
